# Identification of Specific Plasma miRNAs as Potential Biomarkers for Major Depressive Disorder

**DOI:** 10.3390/biomedicines12102165

**Published:** 2024-09-24

**Authors:** Cătălin Prodan-Bărbulescu, Cristian Daniel Alin, Ionuţ Flaviu Faur, Georgeta Cristiana Bujor, Edward Paul Şeclăman, Virgil Enătescu, Alexandra-Ioana Dănilă, Ecaterina Dăescu, Rami Hajjar, Laura Andreea Ghenciu, Paul Tuţac, Paul Paşca, Anca Maria Cimpean, Ciprian Duta

**Affiliations:** 1Faculty of Medicine, “Victor Babeş” University of Medicine and Pharmacy Timisoara, 300041 Timisoara, Romania; catalin.prodan-barbulescu@umft.ro (C.P.-B.); bujor.cristiana@umft.ro (G.C.B.); eseclaman@umft.ro (E.P.Ş.); enatescu.virgil@umft.ro (V.E.); alexandra.danila@umft.ro (A.-I.D.); daescu.ecaterina@umft.ro (E.D.); rami.hajjar@umft.ro (R.H.); bolintineanu.laura@umft.ro (L.A.G.); paul.tutac@umft.ro (P.T.); drpascapaul@gmail.com (P.P.); acimpeanu@umft.ro (A.M.C.); duta.ciprian@umft.ro (C.D.); 2Department I—Discipline of Anatomy and Embryology, Faculty of Medicine, “Victor Babeş” University of Medicine and Pharmacy Timisoara, 2nd Eftimie Murgu Square, 300041 Timisoara, Romania; 3IInd Surgery Clinic, Timisoara Emergency County Hospital, 300723 Timisoara, Romania; 4Doctoral School, “Victor Babes” University of Medicine and Pharmacy Timisoara, Eftimie Murgu Square 2, 300041 Timisoara, Romania; 5Department of General Surgery, “Colţea” Clinical Hospital, “Carol Davila” University of Medicine and Pharmacy, 050474 Bucharest, Romania; 6Faculty of Medicine, “Carol Davila” University of Medicine and Pharmacy, Bulevardul Eroii Sanitari 8, 050474 Bucharest, Romania; 7X Department of General Surgery, “Victor Babes” University of Medicine and Pharmacy, 300041 Timisoara, Romania; 8Department IV—Biochemistry and Pharmacology, Faculty of Medicine, “Victor Babeş” University of Medicine and Pharmacy Timisoara, 2nd Eftimie Murgu Square, 300041 Timisoara, Romania; 9Discipline of Psychiatry, Department of Neurosciences, University of Medicine and Pharmacy Victor Babes Timisoara, 300041 Timisoara, Romania; 10Department III—Discipline of Physiopathology, Faculty of Medicine, “Victor Babeş” University of Medicine and Pharmacy Timisoara, 2nd Eftimie Murgu Square, 300041 Timisoara, Romania; 11Department of Microscopic Morphology/Histology, “Victor Babes” University of Medicine and Pharmacy, 300041 Timisoara, Romania; 12Center of Expertise for Rare Vascular Disease in Children, Emergency Hospital for Children Louis Turcanu, 300011 Timisoara, Romania

**Keywords:** major depressive disorder, miRNAs, major depressive disorder biomarkers, hsa-miR-874-3p, hsa-let-7d-5p, hsa-miR-93-3p

## Abstract

Backround: Depression is a significant concern in clinical and preclinical psychoneurobiological sciences due to its high prevalence and its individual and collective consequences. Identifying efficient biomarkers for accurate diagnosis is crucial, with ideal biomarkers having detectable serum levels and conformational and thermal stability. This study aims to identify stable plasma biomarkers for the diagnosis and prognosis of major depressive disorder, as the pathogenesis of the disorder remains incompletely understood, affecting diagnosis accuracy. Methods: Thus, this study included ten MDD patients and eight healthy controls. The present work analyzed miRNAs in patients with major depressive disorder compared to healthy controls. Results: Eleven specific miRNAs, particularly hsa-miR-874-3p; hsa-let-7d-5p; and hsa-miR-93-3p showed upregulation-type plasma variations in the group of patients with major depressive disorder. miRNA functionality is linked to depressive pathophysiology. Conclusions: This study identifies a “bouquet” of miRNAs with significant upregulation variations in patients with major depressive disorder, suggesting further research to determine their suitability for personalization and evaluation, ultimately becoming integral components of major depression serological evaluations.

## 1. Introduction

### 1.1. Major Depressive Disorder (MDD)

Depression remains a focus for specialized studies in the field of clinical and preclinical psychoneurobiological sciences (Biochemistry, Pharmacology, Pathophysiology, Morphopathology, Psychiatry, Neurology) due to its high prevalence and significant individual and collective levels [1,2]. Depression is a complex mental health disorder with a multifaceted etiology encompassing genetic, environmental, and neurobiological factors [2,3,4,5].

From a historical perspective, depression was originally referred to as melancholia. Melancholia was diagnosed from the early 19th century until the 1880s, leading to the development of the modern concept of depression. The writings of twelve authors, who emphasized the role of faculty psychology and insight, trace the development of this idea [6,7].

Between the 1780s and the 1830s, five writers defined melancholia as a mental illness or deficiency often linked to depression. In the 1850s, Guislain, Bucknill, and Tuke highlighted melancholia without delusions as a neglected disorder [1,2]. By the 1860s and 1880s, five authors concluded that melancholia was a primary mood disorder with delirium naturally resulting from aberrant mood [6,7]. The concept of melancholia included various forms of quiet insanity, with depression becoming more clearly defined in the 19th century as affective disorders. Criteria for diagnostic schemes were introduced in the 1930s, and the modern distinction between unipolar and bipolar disorder emerged in the 1960s. Debates arose over the differences between psychotic and neurotic depression and between endogenous and reactive depression [6,7].

From the point of view of the diagnosis, according to the Diagnostic and Statistical Manual of Mental Disorders (DSM-5), major depressive disorder (MDD) is characterized by distinct changes in cognitive, affective, and neurovegetative functions, and from a temporal point of view, it is defined by episodes lasting at least 2 weeks. From a quantitative point of view, five or more symptoms must be present during the same episode, with at least one being either a depressive state or anhedonia [3,8].

The clinical spectrum may include the following symptoms: significant weight loss or weight gain, fatigue, inability to concentrate, feelings of worthlessness or guilt, and recurrent suicidal thoughts. However, because MDD is a clinically heterogeneous disorder, implementing a symptom-based approach has significant limitations. This is made more difficult by the fact that depression is frequently experienced by individuals suffering from diabetes mellitus, cancer, and neurodegenerative diseases, like Parkinson’s and Alzheimer’s disease [5,9,10].

The DSM-5 categorizes depressive disorder into several distinct subtypes of depression. These subtypes, according to the DSM-5, are labels meant to denote particular symptom patterns or timeframes in which symptoms may manifest [4,9,10,11].

### 1.2. A Genetic Perspective on Major Depressive Disorder

Large-scale consortium and cohort studies have demonstrated that genetics plays a significant role in the etiology of MDD. The pathophysiology of MDD appears to be predominantly influenced by gene–environment interactions, with early-life stress representing a significant risk factor for several mental disorders, including MDD.

With over 1.2 million participants, the largest genome-wide association study (GWAS) on depression found 223 independent single nucleotide polymorphism (SNPs) and 178 genetic risk loci linked to major depressive disorder (MDD). The development of the nervous system, the size of the brain, and the assembly and function of synapses are the top biological processes associated with the approximately 11.3% SNP-based heritability for MDD. Neural growth regulator 1 (NEGR1), CUGBP Elav-Like Family Member 4 (CELF4), and dopamine D2 receptor (DRD2) are among the top genes associated with MDD [12].

Early-life stress causes a persistent disturbance of the hypothalamic–pituitary–adrenal (HPA) axis by acting through epigenetic controls. The epigenetic controls that are significant in MDD are histone modification, non-coding RNA regulation, and deoxyribonucleic acid (DNA) methylation [13,14].

Early-life stress can lead to lifelong increases in glucocorticoid secretion and disruption of homeostatic mechanisms that regulate the activity of the hypothalamic–pituitary–adrenal (HPA) axis, increasing the risk of stress-related diseases like mood and affective disorders, anxiety disorders, borderline personal disorder (BPD), and posttraumatic stress disorder (PTSD). DNA methylation plays a role in translating social experiences into long-lasting changes in gene expression and phenotypes.

The glucocorticoid receptor gene (NR3C1) is a highly conserved transcriptional regulator that controls endocrine responses to stress, metabolism, inflammation, and reproduction. Studies have shown that persistent changes in exon 17 DNA methylation occur as a function of maternal care, and childhood trauma, suicide, BPD, and PTSD produce similar epigenetic modifications [15].

Early-life stress induces site-specific hypermethylation of an nuclear receptor subfamily 3 group C member 1 (Nr3c1) control element, which coordinates the expression of multiple glucocorticoid receptor (GR) transcripts and overall GR protein in Corticotropin-Releasing Hormone (CRH)-expressing neurons and prevents upregulation of CRH under conditions of chronic stress in adulthood. This form of ‘molecular plasticity’ enables an organism’s capacity to mount an adaptive response through the integration of multilayered gene–environment interactions [15].

### 1.3. Implication of miRNA in Major Depressive Disorder

Beginning with the discovery of the first small non-coding RNA in 1993, the knowledge of microRNAs (miRNAs) has expanded rapidly so that recent research has increasingly focused on the role of miRNAs in the pathogenesis of depression [16,17].

miRNAs are a class of small single-stranded RNAs consisting of approximately 18–25 nucleotides and were first identified by Reinhart et al. by developmental time control in *C. elegans* [18].

These small non-coding RNA molecules are, therefore, involved in the posttranscriptional regulation of gene expression and play important roles in various biological processes, including development, differentiation, cell proliferation, and apoptosis [17,19,20,21,22].

The ability of miRNAs to regulate gene expression is a fundamental component of these complex biological processes. Because unique sequences of the RISC/miRNA complex can bind to the 3′ untranslated region of targeted mRNAs, miRNAs are generally thought to cause translational repression or degradation of mRNAs and direct posttranscriptional regulation of genes [11,16,23,24]. Thus, miRNAs play an important role in controlling and regulating cell development and differentiation, and particularly in the brain, their expression can significantly modulate neuronal development and intracellular pathway signaling in apoptosis [5,23].

miRNAs are a type of DNA polymerase that transcribes genomic regions into primary miRNAs. These primary transcripts are processed by a microprocessor complex, Drosha and DGCR8, to yield a 70-nucleotide pre-miRNA [20,24]. Exportin-5 exports the pre-miRNA to the cytoplasm, allowing for it to be transported through the nuclear pore complex. Dicer processes this to yield a mature miRNA duplex. miRNAs are part of the miRNA-induced silencing complex (miRISC), which inhibits transcription and mediates mRNA decay. They have a significant influence beyond intracellular environments, including the nervous system. miRNAs can be released into extracellular fluids, where they circulate systemically to modulate distant target cells. Their transport mechanism involves exosomes, a specialized subtype of extracellular vesicles (EVs), secreted by various cell types [21,23,24].

### 1.4. Pathogenic Mechanisms of miRNA in Major Depressive Disorder

In the context of depression, miRNAs are considered to influence the pathogenesis of the disorder through several mechanisms, such as regulation of neurotransmitter systems, neuroplasticity, stress response and HPA axis regulation, inflammation, and neurogenesis.

Thus, miRNAs can regulate the expression of genes involved in neurotransmitter systems, such as serotonin, dopamine, and glutamate, which are crucial for mood regulation and have been implicated in the pathophysiology of depression [21,23,24]. The mechanism by which miRNAs play an important and relevant role in regulating neurotransmitter signaling is as follows. Neurotransmitters are synthesized in the presynaptic neuron. Subsequently, they are embedded in vesicles and released into the synaptic slit. These vesicles will interact with postsynaptic receptors to stimulate or inhibit postsynaptic cell activity. Every component of this process, from neurotransmitter synthesis to signal termination, is modulated by the activity of synaptic proteins that can be encoded by mRNAs that are targets of miRNA, and some of this regulation can occur locally [21,23].

The hypothalamic–pituitary–adrenal (HPA) axis plays a central role in the body’s response to stress. Dysregulation of the HPA axis has been linked to depression. miRNAs can influence the stress response by regulating the genes involved in the HPA axis and stress-related signaling pathways [25,26].

Increasing evidence suggests that inflammation may contribute to the pathogenesis of depression. miRNAs can regulate the expression of pro-inflammatory and anti-inflammatory cytokines, thus potentially influencing inflammatory processes associated with depression [25,26].

miRNAs are involved in neurogenesis, the process by which new neurons are generated in the brain. Impaired neurogenesis, particularly in the hippocampus, has been associated with depression. miRNAs may affect neurogenesis and, by extension, the pathophysiology of depression [9,25,26,27,28].

An extensive body of research has focused on the fact that miRNA plays a critical role in the pathogenesis of depression (e.g., in neuroplasticity and neurogenesis), resulting in clinically relevant symptoms (e.g., suicidal behavior).

The reason why miRNAs are a current scientific attraction is due to the following characteristics:

(1) They are stably maintained and transported in various biological fluids, e.g., cerebrospinal fluid and peripheral serum/plasma;

(2) The method used for their detection is technically simple and inexpensive;

(3) Their expression is tissue and disease specific.

Another important element is that there are several miRNAs that are reported in the literature as diagnostic or treatment-monitoring elements in major depressive disorder, and these in our study do not show statistically significant variations.

This study aims to identify stable plasma biomarkers for the diagnosis and prognosis of major depressive disorder, as the pathogenesis of the disorder remains incompletely understood, affecting diagnosis accuracy.

## 2. Materials and Methods

### 2.1. Study Sample and Design

This study included 10 patients with major depressive disorder and 8 healthy controls. The subjects included in this study were patients of the Psychiatric Clinic “Eduard Pamfil” in Timisoara, within the County Emergency Hospital “Pius Brinzeu” Timisoara. These patients met the diagnostic criteria for a major depressive episode as outlined in the Diagnostic and Statistical Manual of Mental Disorders, Fourth Edition, Text Revision (DSM-IV-TR).

The sampling method was by convenience, selecting patients who were readily available and met the inclusion criteria. This approach, while not random, was appropriate given the clinical setting and the specific population under study.

In this study, patients were drug-naïve, meaning they had not previously received any pharmacological treatment for their condition. This was crucial for ensuring that baseline evaluations were not influenced by prior medication, providing a clear understanding of the patients’ initial clinical status.

This study was conducted in accordance with the Code of Ethics of the Declaration of Helsinki, which outlines ethical principles for medical research involving human subjects. This ensured that this study upheld the highest standards of ethical conduct, prioritizing patient safety and rights.

This study received approval from our institutional review board (IRB). The IRB approval process included a thorough review of the study protocol, ensuring that the research met ethical and regulatory requirements.

All participants provided signed informed consent. This process involved explaining the study’s purpose, procedures, potential risks, and benefits to the patients. They were assured that participation was voluntary and that they could withdraw at any time without any impact on their clinical care.

Patients participated in this study voluntarily and freely. They were not offered any financial compensation, ensuring that their participation was motivated by a genuine interest in contributing to the research rather than monetary incentives.

The inclusion criteria were the following: age between 18 and 65 years, a positive diagnosis of MDD according to the DSM-IV-TR criteria (eligible patients also underwent a SCID-I Romanian research version of a clinical interview for psychiatric disorders of axis I DSM-IV-TR), and willingness to complete informed consent.

The exclusion criteria were the following: presence of another ongoing psychiatric disorder diagnosed according to axis I of DSM-IV-TR; antidepressant treatment in the last 12 months prior to study enrollment; presence of chronic medication treatment for a comorbid medical condition 12 months before or during the study; and presence of pregnancy during the study.

All subjects considered for this study underwent a general physical examination, and a blood sample was sent to a clinical laboratory for complete blood count analysis.

### 2.2. Profiling of miRNAs

Venipuncture was used to collect blood samples in EDTA-coated vacutainers, and plasma was separated by centrifugation within 2 h of collection and stored at −80 °C until further use.

Total RNA containing miRNA was extracted using the miRNeasy Serum/Plasma kit (Qiagen—Dusseldorf Germany), including *C. elegans* miR-39 mimic spike-in and UniSp2, 4, and 5 as an internal normalization control, according to the manufacturer’s recommendations.

Mature miRNA expression was determined by real-time PCR in an ABI 7900HT System (Thermo, Waltham, MA, USA) using the miRCURY LNA miRNA Focus Panels (Qiagen, Dusseldorf Germany), which interrogates 179 unique human miRNAs, following the manufacturer’s procedures.

Real-time PCR data were analyzed using the online QIAGEN GeneGlobe Data Analysis Center (Qiagen—Dusseldorf Germany).

### 2.3. Statistical Analysis

Statistical analysis, target prediction, and pathway analysis.

Only miRs showing expression in all samples were selected for further analysis. The fold changes in normalized miRNA expression between the two timepoints (before and after treatment) were calculated, and a paired *t*-test was used for statistical significance.

Fold change (2^−ΔΔCT^) is the normalized miRNA expression (2^−ΔCT^) in the test sample divided by the normalized miRNA expression (2^−ΔCT^) in the control sample.

The *p*-values were calculated based on a Student’s *t*-test of the replicate 2^−ΔCT^ values for each gene in the control group and treatment groups, and *p*-values less than 0.05 are indicated in red.

### 2.4. Bioinformatic Analysis and Identification of the Relationships between miRNAs and Potential Pathways from MDD

We used the miRTarBase database (https://mirtarbase.cuhk.edu.cn/~miRTarBase/miRTarBase_2022/php/index.php, accessed on 30 August 2024) for the assessment of the top 5 MDD-related upregulated miRs with the highest fold change values identified in the present study. We had selected target genes with 2 or more verification methods for more confidence. The selected genes were uploaded into the Erichr website (https://maayanlab.cloud/Enrichr/, accessed on 30 August 2024) and MDD-related pathways were selected by using Reactome and KEGG as reference databases [29,30,31]. The resulting pathways were linked with MDD published data. A *p*-value less than 0.05 was considered significant.

## 3. Results

We investigated ten patients with MDD and eight healthy controls. The controls, who varied in age from 18 to 65 years, were the partners of eligible and enrolled study subjects with an age difference of less than five years.

In Figure 1 the general distribution of the studied group is represented, showing the multitude of miRNAs that are upregulated/downregulated in the patient group compared to the normal group.

Going from general principles to specific situations, a set of hsa-miRNAs were detected that showed multiple statistically significant variations, and the first set (being a preliminary study) of these is presented as Table 1.

Following the preliminary analysis, these hsa-miRNAs showed significant fold regulations in the patient group versus the control group. They are considered statistically significant, and some have been identified in the literature as reported by other researchers.

The fact that these hsa-miRs show a well-represented fold regulation with statistical significance means that they could be given a biomarker role in the diagnosis of major depressive disorder. Given that fold change (2^−ΔΔCT^) is the normalized miRNA expression (2^−ΔCT^) from the test sample divided by the normalized miRNA expression (2^−ΔCT^) of the control sample, fold regulation (FR) represents the fold change results in a biologically meaningful way.

Fold change values greater than one indicate positive regulation or upregulation, and FR equals fold change. It can be observed in our study that hsa-miR-93-3p shows an FR = 8.47, which enables it as a viable marker for major depressive disorder. In other words, the viability of this marker was also indicated by the study of Homorogan et al., by performing its assay in blood samples of patients [20].

In the case of Table 1, we performed an analysis of miRNAs showing statistically significant expression in our group of patients with major depressive disorder compared to the healthy group. We also sought to add studies from the literature highlighting the same miRNAs with significant variation in major depressive disorder.

For example, in the case of hsa-miR-874-3p, in our study, this miRNA is upregulated; in the study by Suento et al., hsa-miR-874-3p can be used as a therapeutic target in major depressive disorder in view of the fact that miR-874-3p may play a major role in the prevention of lipopolysaccharide-induced depressive behavior through inhibition of indoleamine 2,3-dioxygenase 1 expression [32]. It can also be observed that hsa-miR-874-3p shows statistically significant changes and an FR = 19.4, the highest upregulation in the test group.

The next hsa-miRNA described is hsa-let-7d-5p, which shows upregulation modulation in the plasma of patients with major depressive disorder, with an FR = 13.45 and a *p* = 0.02. Literature studies show that let-7 family members are highly expressed in the human brain and play an important role in synaptogenesis and neurogenesis [33]. In a meta-analysis led by Maffioletti et al., downregulation of hsa-let-7e-5p expression was associated with major depressive disorder [32].

The following element is represented by hsa-miR-93-3p, which in the present study shows FR = 8.47 and *p* = 0.02, which enables it as a viable marker for major depressive disorder. In other words, the applicability of this marker was also indicated by the study of Homorogan et al., by testing it in blood samples of patients [34].

Therefore, the miRNAs described above (hsa-miR-874-3p, hsa-let-7d-5p, and hsa-miR-93-3p) as well as the others in Table 1, for instance, hsa-miR-574-3p, hsa-miR-132-3p, hsa-miR-125a-5p, hsa-miR-376a-3p, etc., show statistical significance in the studied group, indicating a potential biomarker role in major depressive disorder. The first three elements (hsa-miR-874-3p, hsa-let-7d-5p, and hsa-miR-93-3p) are more representative due to the FR values, as they have the highest values in the studied group, so they are the most upregulated and relevant.

From an objective point of view, the quantitative variations in these hsa-miRs are represented graphically as Figure 2. It is clearly evident that there is a considerable upregulation trend of these miRNAs in the studied group, which reveals their importance as biomarkers, prognostic factors, or therapeutic targets in major depressive disorder.

Bioinformatic analysis performed for the first most relevant upregulated genes with the highest fold regulation values found in the present study helped us to link them not just with the current MDD disease but showed us that these patients may be candidates for other neurological or neurodegenerative diseases. The clinical impact of the present findings may help to identify an MDD patient’s subgroup with a high prognostic significance to develop in the future other nervous system-related disease. A high pathway heterogeneity was identified during analysis of each miR.

For hsa-miR-874-3p, we identified 62 target genes overlapped to major depressive disorder. Enrichment analysis identified seventeen significant pathways (with a *p*-value less than 0.05, Table 2, seven of them being directly related to the major depressive disorders in the literature. For these seven pathways, we identified eleven overlapped genes related to three clusters. According to Figure 3, Cluster 7 included most of the nervous system disorders related to MDD, while cluster 5 was found to be related to the arachidonic acid metabolism pathway and cluster 1 to the thyroid hormones pathways, as is shown in Scatter Plot 1 and Table 3.

The next miRNA with a high fold regulation value was hsa-let-7d-5p. For it, we did not identify a significant potential pathway related to MDD.

Fifty-eight target genes were identified to be associated with hsa-miR-93-3p in the present study. From all these target genes, HHAT and SMURF 2 were associated to Hedgehog signaling pathways, which seems to be the unique significant pathway (*p* = 0.01151) detected in the present study corresponding to Cluster 6 (Figure 4). Although the following pathways had no significance (*p*-value > 0.05), cellular senescence and axon guidance are important pathways related to major depression.

hsa-miR-574-3p had the fourth fold regulation value in our study. From this miR, we detected 25 associated target genes significantly involved in 50 pathways. From these 50 pathways, only the thyroid hormones signaling pathway (based on RXR alpha and EP300 target genes) was found to be overlapped to major depression. We must mention that this pathway was also found to be significant for hsa-miR-874-3p (HDAC, APT2A2, and ESR1 target genes, *p* = 0.009).

The last miR assessed by this analysis was hsa-miR-140-5p. Eighty-seven associated target genes were selected corresponding to 28 significant pathways mostly related to malignant diseases. Only the prolactin secretion pathway (STAT1, ESR2, *p* = 0.03) has been identified as being involved in MDD pathogenesis.

## 4. Discussion

The aim of this study was to analyze a set of miRNAs in patients diagnosed with major depressive disorder compared to healthy controls. This study identified 11 specific miRNAs (hsa-miR-874-3p, hsa-miR-574-3p, hsa-miR-451a, hsa-miR-93-3p, hsa-let-7d-5p, hsa-miR-125a-5p, hsa-miR-132-3p, hsa-miR-141-3p, hsa-miR-140-5p, hsa-miR-376a-3p, hsa-miR-423-3p) that showed multiple plasma variations in patients with major depression.

Following a review of the literature, we found several significant correlations with our study, for example, hsa-miR-93-3p, an miRNA that showed significant variations in our study of patients diagnosed with major depressive disorder.

In a similar manner, in the study conducted by Homorogan et al., the role of hsa-miR-93-3p in major depressive disorder as a diagnostic marker was investigated and highlighted [19].

hsa-miR-93-3p is encoded by a gene located on chromosome 7q221. It co-transcribes with the host minichromosome maintenance complex component 7 (*MCM7*) gene and is expressed in the nucleus. It belongs to the miRNA-106b-25 pro-oncogene cluster and is a paralog of the miRNA-17-92 cluster. This cluster has been shown to control the expression of a number of target genes related to critical physiological functions, such as angiogenesis, apoptosis, and cell division [35]. Furthermore, the study conducted by Liu X. et al. highlighted the role of hsa-miR-93-3p alongside miR-101 as biomarkers for the diagnosis of major depressive disorder [34,35,36].

hsa-miR-874-3p is clearly highly upregulated in our patient group, having considerable diagnostic value in major depressive disorder. According to scientific papers, it is known that hsa-miR-874 is located on chromosome 5q31.2, playing important roles not only in depression but also in malignant diseases (hepatocellular carcinoma, pancreatic ductal adenocarcinoma, epithelial ovarian cancer, breast cancer, colorectal cancer, gastric cancer) and non-malignant diseases (ischemic heart disease, acute myocardial infarction, diabetic nephropathy, and polycystic ovary syndrome) [10,37,38,39,40,41,42,43].

According to Zhang et al., there are many miR-874 target genes, and by inhibiting the expression of these target genes, miR-874 is implicated in cell proliferation, apoptosis, migration, invasion, cell cycle, and epithelial–mesenchymal transition [35].

miRNA let-7 (let-7), one of the first miRNAs identified and characterized, exerts multiple biological roles, being involved in several physiological processes that include signal modulation, temporal regulation, lens regeneration, protein ubiquitylation, sexual identity, and cell proliferation and differentiation [44]. Returning to the realm of neurobiology and neuropsychiatry, more significantly, let-7 family miR-NAs are expressed throughout the brain and are associated with a range of neurophysiological processes, including mood and affective behaviors, as well as multiple psychiatric disorders, such as anxiety, depression, schizophrenia, and cocaine addiction [44,45].

In the study conducted by Maffioleti et al., the presence of hsa-miR-let-7d-5p as a diagnostic biomarker in major depressive disorder is emphasized by showing upregulation-like variations in the plasma of patients with major depressive disorder. Additionally, in the study led by Maffioleti and his working group, other hsa-miRNAs of the hsa-let-7d family, which show upregulation-like variations, are also highlighted, for example, hsa-let-7a-5p and hsa-let-7f-5p [32,33,34,35,36,37,38,39,40,41,42,43,44,45,46].

There are multiple specialized studies that describe variations in the hsa-let-7 family in major depressive disorder, some of which are highlighted as upregulation variations (hsa-let-7e-5p,) and others highlighted as downregulation variations in the hsa-let-7 family (hsa-let-7b-5p, let-7c-5p) [44,45,47,48].

Regarding the association of miRNAs and pathogenic mechanisms in major depressive disorder (inflammation, HPA axis, neurogenesis, neuroplasticity, neurotransmitter regulation, and neuroplasticity), it is important to note that miRNA-let7 shows a particularly important set of actions; for example, Let-7d targets D3R in the hippocampus, Let7b and let7c regulate the PI3k-Akt-mTOR pathway, as proved by studies in the literature [44,45,47,48,49,50,51,52,53].

Another significant aspect concerns hsa-miR-132-3p, one of the eleven miRNAs demonstrating statistically significant variation in this study group and which correlates with the findings from specialized studies. Ortega’s study reveals that hsa-miR-132-3p regulates hippocampal BDNF levels, leading to reduced gray matter volume and poorer cognitive performance. Its role is linked to depression self-assessment scales and HAM-D scales. MiR-132 may contribute to MDD and cardiovascular disease co-occurrence, with suppression increasing BDNF levels [54,55,56,57,58].

Another significant aspect pertains to the utility of these miRNAs as possible “targets”. This method is known as the antagomir concept, where miRNAs, despite their current role as diagnostic, prognostic, or therapeutic biomarkers, can also be used as targets for antagonists [5,26].

MicroRNA functionality is linked to depressive pathophysiology, wherein overexpressed miRNAs silence downstream targets. Antisense technology, specifically antagomirs (cholesterol-conjugated 2-O-methyl RNA antisense oligonucleotides), selectively suppress pathologically abundant miRNA and target aberrant miRNA expression in various disease models, facilitating targeted treatment of depression and further elucidating the role of microRNA in depression [24].

Additionally, nanoparticles represent a significant avenue for present and future applications. According to the study by Bui Duc Tri, nanoparticles, such as GABA nanoparticles, offer multiple mechanisms of action in treating major depression. The GABA released from GABA nanoparticles is rapidly consumed by GABA-T, yet it is simultaneously sufficient to enhance the activity of the GABAergic pathway in the gut. This initial evidence supports the activation of the gut–brain axis as a possible mechanism for the therapeutic effects of GABA nanoparticles in the treatment of MDD [59,60,61,62].

The study by Xiao-Lie He et al. demonstrated that loading solid lipid nanoparticles with curcumin and dexanabinol increased mRNA protein expression levels and protein expression of mature neuronal markers [58]. Furthermore, it promoted the release of dopamine and norepinephrine, thus presenting antidepressant effects in the treatment of major depressive disorder [61]. These studies underscore the significant role of miRNA in major depression treatment.

A special emphasis should be placed on the lipid profile of the depressed patient, as it is known that low serum total cholesterol, HDL cholesterol, and LDL cholesterol levels are associated with suicidal tendencies. This association is attributed to the cholesterol concentration in the cell membrane, where lower levels of cholesterol correlate with decreased 5-HT receptor density, as evidenced by studies [37,40,63].

This study has some limitations that warrant acknowledgement. Firstly, it was conducted on a relatively small group comprising ten patients diagnosed with major depressive disorder and eight healthy controls. Additionally, this study did not categorize the patients with major depressive disorder into specific illness subtypes (e.g., major depressive disorder with psychotic elements or major depressive disorder with melancholic elements), limiting the categorization of results to broad major depressive disorder categories.

The discovery of efficient biomarkers involved in illness development aids in the accurate diagnosis of depression. To discuss the incorporation of a biomarker into the battery of clinical tests performed on patients and individuals at risk of the disease in a serious and evolving manner, an ideal biomarker has the following characteristics: detectable serum level and conformational and thermal stability in biological fluids.

MDD may be considered a highly heterogeneous group of diseases due to its multiple non-neurological and neurological disease involvement in its onset and development. MDD heterogeneous etiology has a direct impact on therapy efficiency, and this is an underscored aspect of the therapeutical approach in MDD. Several subtypes of MDD are recognized to be linked to resistance to conventional therapy [58,59,60,61,62]. There is uncertainty regarding the diagnostic criteria for major depressive disorder (MDD), specifically for treatment-resistant depression (TRD) and partially responsive depression (PRD) [58,59,60,61,62]. These criteria may be related to other diseases, such as endocrine or metabolic ones. Our study identifies two MDD pathways related to patients’ endocrine profile. Several previously published papers reported involvement of the thyroid signaling pathway in the pathogenesis of major depressive disorders [59,60,61,62,63,64,65], but only a few of them recently published have linked this pathway to miRs [62,63,64]. The present paper identified five target genes associated with two miR types for the thyroid signaling pathway: HDAC, ATP2A2, ESR1, RXR alpha, and EP300. The study of HDAC, ATP2A2, and ESR1 highly sustain their involvement in MDD pathogenesis and therapy response. HDAC is already certified as a major factor of stress-induced major depression by epigenetic mechanisms [65,66] but also of therapy response [67]. Our results overlap the data in the literature for HDAC. ATP2A2 gene mutation is specific for Darier disease, a genetic disease characterized by skin lesions associated with neuropsychiatric symptoms, mainly major depression (30%) [65,66,67,68,69]. None of our patients were previously diagnosed with Darier disease, but ATP2A2 was one of the target genes associated with the thyroid signaling pathway but also with the pathways of neurodegeneration in the present study, suggesting a link between these two pathways in the pathogenesis of MDD. Indirect evidence of the interrelation of these two pathways has been reported in the experimental models but not in humans [70,71]. We assume that our findings of the interrelation between these two pathways asre firstly reported in humans in the present study.

ESR family genes are already validated members for major depression in humans with a special focus on ESR2 involvement in major depression in pregnant women [69,72,73,74,75,76]. In our study, ESR1 and ESR2 were related to prolactin secretion pathways for two miRs in the absence of pregnancy. RXR alpha from the thyroid signaling pathway identified for hsa-miR-574-3p has not previously been reported to be related to major depression, and thus, we have no data to compare our results. We just report its presence for our patients with MDD.

Two papers in the literature associate EP300 with major depression [75,77], one of them being also present in Rubinstein–Taybi syndrome, which was absent in our patients.

The Hedgehog signaling pathway was previously less studied in major depression, being linked with neurogenesis defects responsible for future depressive disorders [78] or as a predictive biomarker for MDD [77,79,80,81,82]. The HHAC gene was related to MDD in only one paper in the literature in patients with schizophrenia, suggesting that MDD may be an atypical form of schizophrenia [77,82], while SMURF2 has not previously been reported as an MDD-related gene.

In summary, this set of miRNAs, the “miRNome” of major depression, encompasses several miRNAs that exhibit quantitative variations in the plasma of patients with major depressive disorder. The miRNome is termed as such due to the human genome (miRbase v22 database) annotating approximately two thousand unique miRNAs, most of which are expressed in the brain, some with brain-specific functions. These findings underpin the theory that stress and trauma may disrupt the regulatory function of miRNAs, significant for maintaining central nervous system homeostasis [23,24].

This study highlights a cohort of miRNAs (hsa-miR-874-3p, hsa-miR-574-3p, hsa-miR-451a, hsa-miR-93-3p, hsa-let-7d-5p, hsa-miR-125a-5p, hsa-miR-132-3p, hsa-miR-141-3p, hsa-miR-140-5p, hsa-miR-376a-3p, hsa-miR-423-3p) displaying statistically significant variations in the study group, corroborated by other researchers in the literature. However, further studies are essential to customize and evaluate these biomarkers comprehensively, gradually integrating them into the panel of elements examined during serological assessments of patients with depressive disorders.

Proceeding from theoretical to practical medicine, the key findings in this study include the necessity of incorporating blood tests from molecular medicine into the treatment of patients with major depressive disorder. This involves providing labs with RT-PCR equipment, educating healthcare staff, introducing the miRNAs package for diagnosis and monitoring, linking specific miRNAs to subtypes of depression, and using antagomir therapy as targeted treatment. New health programs focusing on personalized medical practices are also recommended.

## 5. Conclusions

The present study successfully identifies a “bouquet” of miRNAs, primarily represented by hsa-miR-874-3p, hsa-let-7d-5p, and hsa-miR-93-3p, which exhibit significant upregulation in patients with major depressive disorder (MDD) within the study group. These findings highlight the potential of hsa-miR-874-3p, hsa-let-7d-5p, and hsa-miR-93-3p as true plasma biomarkers for MDD, suggesting their pivotal role in the pathophysiology of this disorder.

Our results underscore the necessity of standardizing protocols to incorporate these miRNAs into clinical practice, emphasizing their potential utility in both the diagnosis and therapeutic follow-up of MDD patients. Such standardization could pave the way for more accurate and personalized approaches to managing MDD, ultimately enhancing patient outcomes.

However, the promising nature of these biomarkers also calls for further research. Future studies should focus on validating the clinical applicability of these miRNAs across diverse populations and clinical settings. Additionally, detailed evaluations are required to explore the potential of these biomarkers in the personalization of treatment strategies, ensuring that therapeutic interventions can be tailored to the unique biological profiles of individual patients.

In summary, hsa-miR-874-3p, hsa-let-7d-5p, and hsa-miR-93-3p represent promising biomarkers for major depressive disorder. Their integration into clinical practice, through standardized diagnostic and therapeutic protocols, holds significant promise for advancing the precision and effectiveness of MDD management. Further research is essential to fully realize their potential and to establish their role in personalized medicine.

## Figures and Tables

**Figure 1 biomedicines-12-02165-f001:**
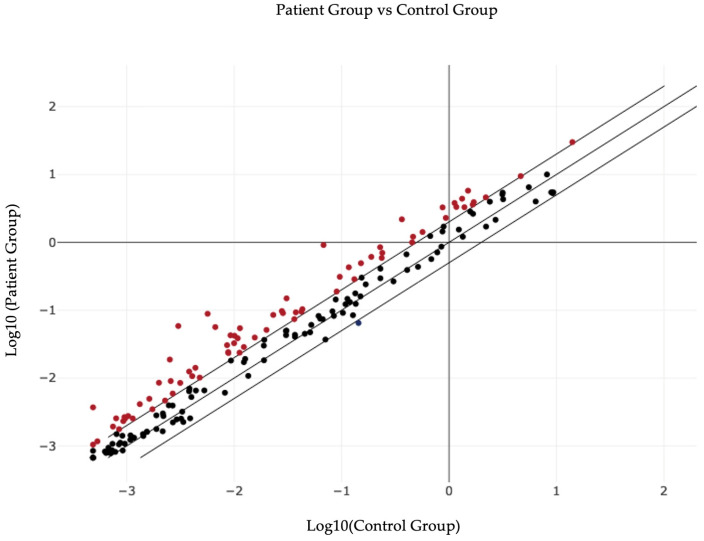
Overall graphical representation of plasma miRNA variations in patient group versus control group.

**Figure 2 biomedicines-12-02165-f002:**
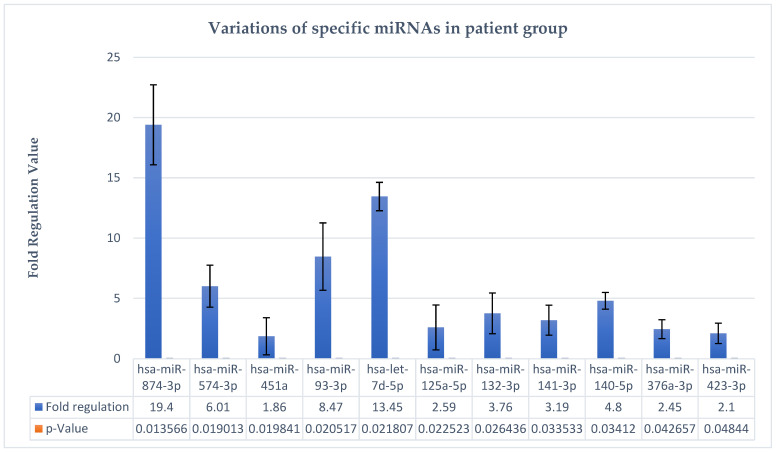
Variations in specific hsa-miRNAs in major depressive disorder.

**Figure 3 biomedicines-12-02165-f003:**
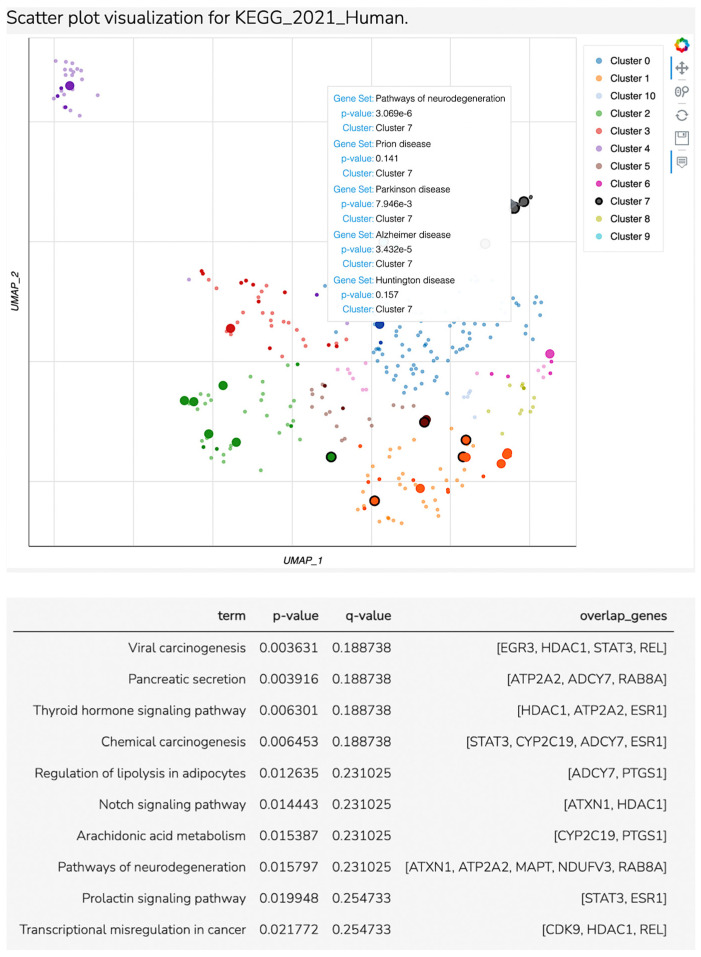
Scatter plot visualization for KEGG_2021 Human.

**Figure 4 biomedicines-12-02165-f004:**
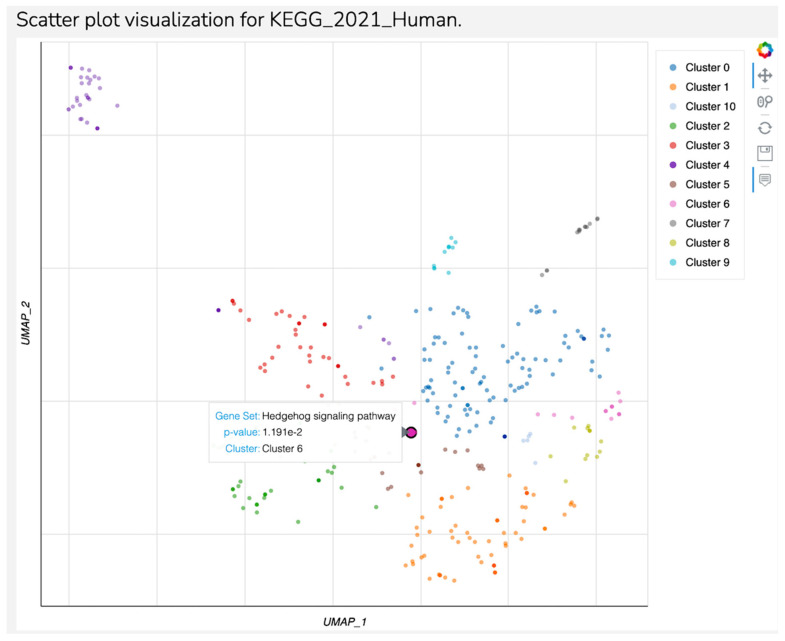
Scatter plot visualization for KEGG 2021 Human—Continuation.

**Table 1 biomedicines-12-02165-t001:** The set of miRNAs specific for major depressive disorder in the study group.

Number	miRNA ID	Fold Regulation	*p*-Value
1	hsa-miR-874-3p	19.4	0.013566
2	hsa-miR-574-3p	6.01	0.019013
3	hsa-miR-451a	1.86	0.019841
4	hsa-miR-93-3p	8.47	0.020517
5	hsa-let-7d-5p	13.45	0.021807
6	hsa-miR-125a-5p	2.59	0.022523
7	hsa-miR-132-3p	3.76	0.026436
8	hsa-miR-141-3p	3.19	0.033533
9	hsa-miR-140-5p	4.8	0.03412
10	hsa-miR-376a-3p	2.45	0.042657
11	hsa-miR-423-3p	2.1	0.04844

**Table 2 biomedicines-12-02165-t002:** Table of top 10 significant *p*-values and q-values for KEGG 2021 Human.

Index	Name	*p*-Value	Adjusted *p*-Value	Odds Ratio	Combined Score
1	Thyroid hormone signaling pathway	0.006023	0.1782	8.69	44.42
2	Pathways of neurodegeneration	0.01480	0.2182	3.70	15.58
3	Arachidonic acid metabolism	0.01492	0.2182	11.42	48.03
4	Prolactin signaling pathway	0.01935	0.2441	9.91	39.08
5	Transcriptional misregulation in cancer	0.02086	0.2441	5.41	20.92
6	Alzheimer disease	0.02611	0.2777	3.76	13.72
7	Serotonergic synapse	0.04666	0.3211	6.06	18.56

**Table 3 biomedicines-12-02165-t003:** Hedgehog pathway is the unique significant pathway detected in the present study, but cellular senescence- and axon guidance-related genes may be linked to major depression despite the lack of significance.

Index	Name	*p*-Value	Adjusted *p*-Value	Odds Ratio	Combined Score
1	Hedgehog signaling pathway	0.01151	0.5293	13.15	58.72
2	Maturity-onset diabetes of the young	0.07277	0.5293	13.98	36.63
3	Cellular senescence	0.07528	0.5293	4.59	11.87
4	Axon guidance	0.09785	0.5293	3.92	9.11

## Data Availability

Data are contained within the article.

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
