# Peer review of "Identification of Specific Plasma miRNAs as Potential Biomarkers for Major Depressive Disorder"

_biomedicines, 2024, doi:10.3390/biomedicines12102165_

Round 1
Reviewer 1 Report
Comments and Suggestions for Authors
This article aims to investigate how TDD diagnosis can be achieved using miRNA as biomarkers. The introduction, although very descriptive, includes data that may not be necessary and delves extensively into various types of depression in sections that lack utility and references. However, it does not update or adequately develop the utility of miRNA for diagnosis and the background of this topic.
The materials and methods section is not very clear or explicit, and the cases are too few to support the conclusions drawn. The discussion is not up to date, with existing works on the topic neither cited nor discussed. Additionally, there are typographical and spelling errors throughout the text.
Author Response
Please see the attachments below.

Reviewer 2 Report
Comments and Suggestions for Authors
Type of article: Article
Journal: Biomedicines
Title of article: Identification of Specific Plasma miRNAs as Potential Biomarkers for Major Depressive Disorder
In this observational study, the authors Dr. rodan-Bărbulescu and colleagues aimed in identifying stable plasma biomarkers, notably microRNAs, for the diagnosis and prognosis of major depressive disorder. The study was conducted with a small sample size, including only 10 patients and 8 healthy individuals, which limits the robustness and generalizability of the data as well as study conclusions. This is a limitation that should be take into account, while additional studies with a larger sample size are necessary. However, several miRNAs, including hsa-miR-874-3p; hsa-let-7d-5p; hsa-miR-93-3p were identified as upregulated patients with major depressive disorder, suggesting their implication in the etiopathogenesis of this disease
Main comment(s).
1. The introduction requires significant reorganization, as it is currently challenging to read due to its verbosity and occasional deviation from the main topic. I strongly recommend reducing its length to sharpen the focus on the primary subject of this work, which is the investigation of circulating microRNAs in patients diagnosed with major depressive disorder. By streamlining the content, the introduction can more effectively guide the reader towards understanding the relevance and significance of this study. Multiple statements and observations belong to the discussion, e.g., lines 279-283. Moreover, I strongly discourage the usage of subhead sections or figures in the introduction, thereby worsening the readability. Conclude the introduction with the aim of the study.
2. Table 1 and Figure 3 present redundant information, making them unnecessary in their current form. To enhance the clarity and conciseness of the work, I recommend removing one of these elements.
3. In order to improve the readability of the manuscript, the discussion should be shortened by 20%
minor observations:
1. Please include the number of patients enrolled in the study in the abstract.
2. The following supporting references concerning the general mechanism of action microRNAs and biogenesis should be included PMID: 36477874, PMID: 35408829 and PMID: 30123182 (lines 203-229)
3. The number of patients and healthy individuals should be clearly stated in the methods (section 2.1). Please also avoid bullet points for a better reading. Also in the methods, please provide supporting references for the methodologies and statistical analyses performed.
4. Plese include a rationale for the selection of the LNA miRNA Focus Panels. Is this mirna panel of particular interest for this disease?
5. Additional studies that investigated circulating non-coding RNAs in patients affected by Major Depressive Disorder have been published. They should be mentioned in the discussion. PMID: 36925948, PMID: 34521053 and PMID: 34521053
Comments on the Quality of English Language
English Language is fine
Author Response
Please see the attachments below.

Reviewer 3 Report
Comments and Suggestions for Authors In the current work, Prodan-Bărbulescu and colleagues investigated the expression level of a panel of miRNAs in patients with major depressive disorder (MDD) and healthy volunteers. Using the miRCURY assay, the authors identified a set of differentially expressed miRNAs (11 miRNAs, including hsa-miR-874-3p, hsa-let-7d-5p, and hsa-miR-93-3p as the most sensitive to the development of MDD) and provide detailed information on their likely involvement in the pathogenesis of depression according to published data. The data obtained provide novel and valuable information for a deeper understanding of the mechanism of MDD development. The manuscript can be published in Biomedicines after a major revision. Major comment: Considering the high impact factor of the journal and its inclusion in the first quartile according to Scimago, the evaluation of miRNA levels alone is not sufficient for publication. Dear authors, please include a bioinformatics block in your study. For example, you can use the MirTarBase database to identify experimentally verified mRNA targets of identified miRs, followed by their functional annotation using ToppGene or Enrichr tools. This analysis can provide independent confirmation of the association of the revealed miRs with mental disorders and additional information about the mechanisms of their development. Minor comments: Unfortunately, a large number of typos were found in the article, which need to be corrected. For example: Line 180 - please add a dot at the end of the sentence. Lines 201, 230, 259, 388, 394, 442, 445, 449, 455, 458 and throughout the manuscript - please add a space before the reference. Line 249 - Delete the extra space between [22] and the dot. Line 303 - Add a space between board and (IRB). Line 333 - What does miR-39miRNA mean? Please correct. Line 344 - Treatment, but not treat- ment. Please correct. Line 366 - Please add space between They and are. Line 379 - The table name should be above the table, not below. Line 398 - FR = 8.47 and p = 0.02. Please correct. Line 402 - The authors may have meant Table 2, not Table 3. Please correct. Lines 400, 434, 499, 543 - please change dot with space before reference. Figure 3 - Please add bars representing standard deviations. Line 428 and throughout the manuscript - please write gene names in italics. Line 432- Liu X et al. Please correct. Line 438, 440, 502 - Please add spaces before parentheses. Line 449 - What does [STAT3 mean? Please correct. Line 472 - Please change comma to dot after [40]. Line 487 - Dear authors, is the name hsa-miR-132-132 spelled correctly? Maybe you mean hsa-miR-132-3p? Please correct or comment. Line 508 - please add dot between depression and The GABA. Line 516 - please add space between presence and mRNA. Line 518 - Please correct itpromoted. Line 525 - Please correct [32-34[36]]. Line 531 - Please decapitalize limiting.
Author Response
Please see the attachments below.

Round 2
Reviewer 1 Report
Comments and Suggestions for Authors
The article has been significantly revised and improved, making it suitable for publication in its current form. However, the sample size remains limited, and the results are somewhat inconclusive.
Author Response
Please see the attachments below.

Reviewer 3 Report
Comments and Suggestions for Authors I respect and thank the authors for making changes according to my minor comments. I realize that the bioinformatics analysis fell during the summer period when most of the staff is on vacation. I suggest that you perform a relatively simple analysis to confirm the association of the identified differentially expressed miRNAs with processes involved in the pathogenesis of major depressive disorder (MDD). To do this, you can follow a series of the following simple steps: (1) Go to the miRTarBase database website (mirtarbase.cuhk.edu.edu.cn/), select Human from the Search --> Species tab, and enter the identified top 5 MDD-related miRs with the highest fold change values one by one. (2) In the table that appears, copy all target genes (Target column) into a separate Excel file, and select target genes with 2 or more verification methods for more confidence (pay attention to the checkboxes). (3) Then go to the Erichr website (maayanlab.cloud/Enrichr/enrich), upload the collected target genes to the "Drop a file or paste a set of Entrez gene symbols..." window, click Submit --> Pathways, select the database of interest (e.g. Reactome or KEGG) and analyze which processes and pathways appear. Try to link the resulting pathways to MDD using published data. Also, please correct the bars representing standard deviations in the corrected Figure 3 - this is clearly an error as the bars are identical and on the same level in the graph.
Author Response
Please see the attachments below.

Round 3
Reviewer 3 Report
Comments and Suggestions for Authors
I am grateful to the authors for their attention to my comments and recommendations. I hope that this paper will stimulate further interest of Dr. Prodan Barbulescu Catalin Flavius and his colleagues in bioinformatics methods, which may significantly deepen their important and interesting studies. The current version of the manuscript is almost ready for publication, however, minor problems should be addressed:
Dear authors, please pay attention to Figure 3. The bars need to be corrected - in the current version, all bars are the same length and placed at the same level, indicating a clear error.
Author Response
Please see the attachments below.
